# Repurposing Simvastatin in Parkinson’s Disease Model: Protection Is throughout Modulation of the Neuro-Inflammatory Response in the *Substantia nigra*

**DOI:** 10.3390/ijms241310414

**Published:** 2023-06-21

**Authors:** Moisés Rubio-Osornio, Carmen T. Goméz-De León, Sergio Montes, Carmen Rubio, Camilo Ríos, Antonio Monroy, Jorge Morales-Montor

**Affiliations:** 1Departamento de Neuroquímica, Instituto Nacional de Neurología and Neurocirugía, Tlalpan, Ciudad de Mexico 14269, Mexico; 2Departamento de Inmunología, Instituto de Investigaciones Biomédicas, Universidad Nacional Autónoma de México, Coyoacán, Ciudad de Mexico 04510, Mexico; 3Unidad Académica Multidisciplinaria, Departamento de Farmacología, Universidad Autónoma de Tamaulipas, Reynosa Tamaulipas 88740, Mexico; 4Departamento de Neurofisiología, Instituto Nacional de Neurología y Neurocirugía, Tlalpan, Ciudad de Mexico 14269, Mexico; 5Dirección de Investigación, Instituto Nacional de Rehabilitación, Tlalpan, Ciudad de Mexico 14389, Mexico; 6Laboratorio de Neuroprotección, Facultad de Farmacia, Universidad Autónoma del Estado de Morelos, Cuernavaca 62209, Mexico

**Keywords:** Parkinson’s disease, MPP^+^, neuroinflammation, oxidative stress proinflammatory cytokines

## Abstract

Parkinson’s disease is a neurodegenerative disorder characterized by oxidative stress and immune activation in the nigro-striatal pathway. Simvastatin regulates cholesterol metabolism and protects from atherosclerosis disease. Simvastatin-tween 80 was administered 7 days before sterotaxic intrastriatal administration of MPP^+^ (1-methyl-4-phenylpyridine) in rats. Fluorescent lipidic product formation, dopamine levels, and circling behavior were considered damage markers. Twenty-four hours and six days after, the animal group lesioned with MPP^+^ showed significant damage in relation to the control group. Animals pretreated with simvastatin significantly reduced the MPP^+^-induced damage compared to the MPP^+^ treated group. As apoptosis promotes neuroinflammation and neuronal degeneration in Parkinson’s disease, and since there is not currently a proteomic map of the nigro-striatum of rats and assuming a high homology among the identified proteins in other rat tissues, we based the search for rat protein homologs related to the establishment of inflammation response. We demonstrate that most proteins related to inflammation decreased in the simvastatin-treated rats. Furthermore, differential expression of antioxidant enzymes in striated tissue of rat brains was found in response to simvastatin. These results suggest that simvastatin could prevent striatal MPP^+^-induced damage and, for the first time, suggest that the molecular mechanisms involved in this have a protective effect.

## 1. Introduction

Parkinson’s disease (PD) is a neurodegenerative disorder characterized by degeneration of the nigrostriatal dopaminergic pathway; however, the causal factor is not yet clearly understood. PD is the second most common neurodegenerative disorder. It is clinically characterized by resting tremor, rigidity, bradykinesia, and postural instability. The main factors involved in neuronal death are an increase in iron content, mitochondrial dysfunction, the over-production of free radicals, and diminished antioxidant response, among others [1]. PD is a multifactorial disorder covering genetic and environmental factors [2]. Accordingly, PD is a disorder that impacts the quality of life in patients, and the lack of curative treatment means that symptoms are difficult to control, affecting predominantly men over women (1.5:1) in people over 65 years old [3]. Currently, there are no pharmacological treatments that slow the progression of the disease or reduce the mortality rate of the remaining neurons. 

Administration of MPP^+^ (1-methyl-4-phenyl pyridine), the active metabolite of neurotoxin MPTP (1-methyl-4-phenyl-1,2,3,6-tetrahydropyridine), reproduces the main neurochemical characteristics of PD in laboratory animals. The MPP^+^ neurotoxic potential is based on its ability to inhibit complex I of the mitochondrial electron transport chain (METC), inducing energy depletion, the over-production of free radicals, oxidative stress, and metal dyshomeostasis [4]. Moreover, previous data indicate a relation between MPP^+^-induced microglial activation and the degeneration of dopaminergic neurons. A PC12 cellular line treated with MPP^+^ induced increases in the mRNA and protein levels of interleukin-6, IL-1β, and tumor necrosis factor-alpha [5]; likewise, MPP^+^ activates the NLRP3 inflammasome in microglia, and the NLRP3 inflammasome-activated microglia plays a pivotal role in the neurodegeneration associated with PD [6].

Simvastatin is an inhibitor of hepatic 3-hydroxy-3-methylglutaryl-coenzyme A (HMG-CoA) reductase—the major rate-limiting enzyme in cholesterol synthesis. Similar to other members of the “statins,” simvastatin lowers total serum cholesterol and particularly low-density lipoprotein (LDL) cholesterol concentrations, thereby reducing the risk of atherosclerosis and its complications. Until now, there is enough accumulated evidence of the effect of simvastatin in some murine models of Parkinson’s disease. 

In this sense, the aim of this study is to describe the effect of simvastatin against MPP^+^-evoked oxidative and inflammatory damage, since we consider that the protein expression map can clarify the differential expression profile induced by simvastatin and its effect on the neurotoxicity of MPP^+^. The possible use of simvastatine as a repurposing drug or as a coadjutant therapy for the dopaminergic neuron preservation in patients with Parkinson’s disease remains to be tested in clinical studies.

## 2. Results

### 2.1. Fluorescent Lipidic Products

Figure 1 shows the results of the effect of simvastatin pre-administration schedule (40 mg/kg/day), which produced no statistical changes in the striatal formation of fluorescent lipidic products (23.4 ± 3.25 Fluorescent Units (FU)/mg protein) versus the control group (17.2 ± 2.88 FU/mg protein). Whereas, MPP^+^-infusion statistically increased (*p* < 0.0001) the formation of lipid fluorescents products (40.1 ± 4.18 FU/mg protein); nonetheless, in the experimental group, the MPP^+^ damage was reversed (*p* < 0.007) via simvastatin treatment (25.3 ± 2.7 FU/mg protein).

### 2.2. DOPAMINE (DA) Content in Striatal Tissue

The group of animals treated via both the oral pathway and intracerebral surgery using simvastatin and sterile saline, respectively, was considered the control group. Basal levels of DA were 61.6 ± 3.91 μmol/g wet tissue. The simvastatin pre-administration schedule with 40 mg/kg/day did not affect striatal DA levels (65.8 ± 6.5 μmol/g wet tissue). In contrast, after microinjection of MPP^+^ (10 μg/8 μL), a significant decrease (*p* < 0.001) was found in DA concentrations (20.6 ± 3.1 μmol/g wet tissue), whereas the simvastatin pre-administration schedule for MPP^+^-treated rats produced a significant (*p* < 0.03) preservation of DA content (47.3 ± 8.4 μmol/g wet tissue) (Figure 2).

### 2.3. Neurodegenerative Damage 

MPP^+^-induced damage significantly (*p* = 0.001) reduced the positive signal for tyrosine hydroxylase (TH) compared to the control group (vehicle plus saline solution). This damage was significantly reduced (*p* = 0.0053) by the subchronic administration of 40/mg/kg when we compared the effect of simvastatin with the control group. Systemic administration of simvastatin and the intrastriatal administration of saline solution presented similar levels of positive signal for TH when compared to the control group (Figure 3).

### 2.4. Circling Behavior

Approximately 6 days after MPP^+^ intrastriatal infusion, we observed a marked effect on circling behavior after apomorphine administration (151.1 ± 18.3 turns/60 min), which was significantly increased (*p* < 0.0007) in comparison with the control groups (vehicle and saline; simvastatin and saline), as shown in Figure 3. Simvastatin pretreatment schedule statistically reduced (*p* < 0.008) the number of turns induced by (45.5 ± 21.3 turns/60 min) MPP^+^ (Figure 4).

### 2.5. Protein Expression Patterns from Striatum from Control, Simvastatin, MPP^+^, and Simvastatin-MPP^+^-Treated Rats by 2D-SDS-Page

Representative gel images of the four experimental groups are shown in Figure 1, which depicts (A) Control group without treatment, (B) Group treated with Simvastatin, (C) Group treated with MPP^+^, and (D) Group treated with Simvastatin and MPP^+^. In a previous report, we found that SMV exerted a protective effect in the rat Parkinson model that seemed to be mediated by an alteration in the cytokine profile in the Nigro striatum. To dissect differences in protein expression, a 2-DE proteomic analysis was performed in the same section of the brain (Figure 5). As depicted in Figure and Table 1, a different protein profile was found for each experimental group. The initial analysis showed that the total number of spots decreased in the Simvastatin/MPP^+^ treated samples, as could be deduced by the number of total spots of 2D gels, while the group exposed to Simvastatin alone exhibited the greatest number of spots (Figure 5 and Table 1).

### 2.6. Bioinformatic Analysis

The neurotoxin MPP^+^ reproduces most of the biochemical and pathological hallmarks of Parkinson’s disease, such as toxicity and therefore inflammation by oxidative stress inducers and production of ROS by activation of the NADPH-oxidase complex. Some proteins that are involved in MPP^+^ inflammation response that were identified in the group injured by MPP^+^ were Platelet-derived growth factor D (Pdgfd), Insulin-like growth factor 1 (Ilgf), IL-1β, GPX1, and Interferon alpha-1 (IFN-α1) (Table 2 and Figure 6).

As apoptosis promotes neuroinflammation and neuronal degeneration in neurodegenerative pathologies such as Parkinson’s disease, we focused the search on proteins that are related to the establishment of inflammation responses (Figure 6 and Table 2). Platelet-derived growth factor D (Pdgfd) is a protein involved in leukocyte migration and, more interestingly, is involved in cell survival via ERK pathway and indirectly by the brain-derived neurotrophic factor pathway. This protein reduced its expression in rats that were treated with simvastatin.

Insulin-like growth factor I (Igf1) is a protein involved in the positive regulation of T cell proliferation. It positively regulates IL-4, Jun, and Bcl-2 pathways while negatively regulating TNF, IL-1β, Casp3, Bax, and NFκb1. This protein reduced its expression in rats that were exposed to the MPP^+^ toxin. TGF-β3 and TGF-β1 exhibited a similar profile expression. TGF-β1 is involved in the positive regulation of glial cell differentiation, and both are involved in the regulation of inflammatory response (Table 2, and Figure 6). Additionally, TGF-β1 has reportedly presented a protective action against MPP+ injury in the rat model. MPP^+^ resulted in a reduction in TGF-β1 production in the substantia nigra and primary VM neurons and microglia [7]. 

Allograft inflammatory factor 1 (Aif1) is involved in the positive regulation of leukocyte migration. S100A8 has a role in leukocyte aggregation. IL-1α is involved in the inflammatory response. Tumor necrosis factor ligand superfamily member 6 (Faslg) is involved in the death of T cells. IL-6r, a part of the receptor for interleukin 6, binds to IL6 with low affinity but does not transduce a signal. Signal activation depends on the association with IL6ST. Activation may lead to the regulation of the immune response, as well as acute-phase reactions. Carcinoembryonic antigen-related cell adhesion molecule 1 (Ceacam1), a protein involved in negative the regulation of immune effector processes, is overexpressed when rats are treated with simvastatin.

IL-1β was detected in the Simvastatin and MPP^+^ groups; surprisingly, the higher expression was exhibited by the simvastatin groups. Sequestosome-1(Sqstm1) is overexpressed in the simvastatin group. This protein is involved in the regulation of I-kappa B kinase/NF-kappa B signaling and is involved in the regulation of apoptosis triggered by inflammatory cytokines. Caspase-6 is involved in the activation cascade of caspases responsible for apoptosis execution. This protein reduces its expression in the MPP^+^/Simvastatin group. Finally, IFN-α1 expression is reduced in the MPP^+^ group.

In a previous report, we demonstrated that EB-pretreated rats injured with MPP^+^ toxin increase PON2 expression at a similar level to that shown by the control, while SOD2 expression was increased in EB, M, and EB/M groups, compared to the control. SOD2 expression was virtually absent in the control group without treatment. In the present study, we found a similar pattern of SOD2 protein, which was highly expressed in the MPP^+^ group and present in the simvastatin group, but undetectable in the control group and the Simvastatin/MPP+ groups. Catalase expression was detected only in the simvastatin group, and GPX1 was overexpressed in the simvastatin group, a little higher in the MPP^+^ compared to the control but absent in the simvastatin/MPP^+^ group (Table 3).

Comparing the expression of proteins that participate in the inflammatory response of MPP^+^ and MPP^+^/Simvastatin groups (Figure 6), we proposed that MPP^+^ damage establishes an inflammatory response in the substantia nigra that induces the secretion of IL1β, a proinflammatory cytokine that stimulates, among others, the secretion of TNF (another proinflammatory cytokine). The activation of both cytokines may be partially regulated by the Sequestosome-1 protein, which is also involved in the autophagy of peroxisomes in response to reactive oxygen species (ROS); these ROS could be regulated by SOD2 and Gpx1. The inflammatory environment could lead to the activation of caspase 6, which promotes the apoptosis regulated by caspases. Probably as a response, IGF1 increases its expression to down-regulate apoptosis, while TGFβ3 could be pro- or anti-inflammatory depending on the environment. Platelet-derived growth factor D recruit macrophages. All these proteins reduce their expression in the group that received simvastatin after MPP^+^, in comparison with the group that only was injured with MPP^+^ (Figure 6).

## 3. Discussion

Parkinson’s disease is a neurodegenerative disorder characterized by an imbalance in transition metals (iron, copper, manganese, and zinc), decreased activity of complex I of the mitochondrial electron transport chain, reduced content of GSH, overproduction of free radicals, and oxidative stress [1,8]; moreover, the stress of the aging process promotes chronic low-level inflammation in the brain [9]. Currently, diverse treatments and therapies are being used to control the clinical symptoms of Parkinson’s disease; however, to date, no pharmacology treatment cures the illness, nor are there treatments that can preserve the remnant dopaminergic neurons or other drugs that can increase neuron half-life in the nigro-striatal pathway, i.e., we do not have drugs that can stop the progression of the disease.

Simvastatin is a long-established hydroxy-methylglutaryl coenzyme A (HMG-CoA) reductase inhibitor. At the maximal recommended dose, it produces an average reduction in low-density lipoprotein cholesterol (LDL-C), accompanied by strong reductions in LDL-C, triglycerides, and apolipoprotein B, and a modest increase in high-density lipoprotein cholesterol [10]. The present study evaluated the neuroprotective effect of simvastatin against MPP^+^-induced neurotoxicity. MPP^+^ (1-methyl-4-phenyl pyridine), the active metabolite of neurotoxin MPTP (1-methyl-4-phenyl-1,2,3,6-tetrahydropyridine), is a molecule that reproduces the main neurochemical alterations observed in Parkinson’s disease [11]. The effect of simvastatin on MPP^+^-induced damage has been proven for a long time [12]. 

According to the findings of this study, a simvastatin dose (40 mg/kg/day) showed a significate antioxidant effect from the order of lovastatin administrated in a dose of 5 mg/kg [13]. Even when this dose of Simvastatin is too high to be translated to human studies, it should be noted that Simvastatin is used only to test the mechanisms of possible neuroprotection against the known damaging mechanisms produced by MPP^+^, such as oxidative stress and inflammatory response. On the other hand, the MPP^+^ model of neurotoxicity is also produced with a single, high dose of the toxicant, while Parkinson’s disease is developed after years of neurodegeneration. Then, what is being modeled in the present study is the mechanism of neuronal death produced by MPP^+^ rather than Parkinson’s disease itself. The translational value of the results presented here is limited to explore the mechanisms of simvastatin neuroprotection against MPP^+^-induced neuronal death. The dose employed to prevent such damage produced higher plasma concentrations when administered to rats [14] than those attained in human studies at high doses [15]. Thus, one limitation of our work is a lack of a direct application to translational human studies. Another aspect of simvastatin in the central nervous system showed its effect against the LPS-induced inflammatory damage and not in the oxidative damage produced by MPP^+^ [16]. However, in this work the effect of simvastatin was effective upon three damage markers: lipid peroxidation, dopamine content, and circling behavior. The first two were evaluated as short-term, and the last was considered a long-term endpoint damage marker. Consequently, the neuroprotective effect of simvastatin was made evident by a broad temporal spectrum of neuroprotection against MPP^+^-induced neurotoxicity, since it was able of counteract the oxidative damage exerted by MPP^+^ during the first 24 h—characterized by oxidative damage preferably. 

In a previous report, we demonstrated that estradiol benzoate (EB) pretreated rats injured with MPP^+^ toxin experience increases in PON2 expression at a similar level to that shown by the control [17], while SOD2 expression increased in EB, MPP^+^, and EB/MPP^+^ groups, with respect to the control, and SOD2 expression was virtually absent in the control group without treatment. In the present study, we found a similar pattern of SOD2 protein, which was highly expressed in the MPP^+^ group and present in the simvastatin group, but undetectable in the control group and the simvastatin/MPP^+^ groups. 

Likewise, catalase expression was detected only in the simvastatin group, and GPX1 was overexpressed in the simvastatin group—a little higher in the MPP^+^ compared to the control, but absent in the simvastatin/MPP^+^ group (Table 3), which is the cause of simvastatin’s clear effect against MPP^+^-induced oxidative stress. MPP^+^-evoked nitrosative damage is manifested mainly from the first 48 to 72 h. This effect is characterized by the overexpression of inducible nitric oxide synthase (iNOS) Ca^2+^-independent [8,18]. Additionally, it becomes clear that simvastatin protected against MPP^+^-evoked neuroinflammation in the long-term, and this, along with the findings described above, was indirectly observed through circling behavior. 

In the same way, it has been shown that simvastatin prevents the inflammatory process and the dopaminergic degeneration induced by the intranigral injection of LPS through the decrease in the induction of interleukin-1beta (IL-1β), tumor necrosis factor-alpha (TNF-α), and nitric oxide synthase (iNOS) [19]; decreasing astrocytes activation as well [20]. This effect of simvastatin is associated with its ability to modulate the N-methyl-D-aspartic acid (NMDA) receptor and reduce 6-hydroxydopamine (6-OHDA) damage [21]. Furthermore, simvastatin reduces the mRNA and protein levels of the N-methyl-D-aspartic acid receptor 1 (NMDAR1) and IL-6 in PC12 cells stimulated with 6-OHDA [22]. 

Additionally, it has been indicated that simvastatin mediates a protective effect on dopaminergic neurons in the substantia nigra compacta in the LPS-PD model, possibly by promoting neuronal repair and regeneration through the induction of brain-derived neurotrophic factor (BDNF) expression and by inhibiting oxidative stress, thus improving substantia nigra function. However, there is strong evidence to suggest that ERK1/2-mediated modulation of the antioxidant system after simvastatin treatment may partially explain the antioxidant activity in experimental parkinsonian models [21].

On the other hand, it has been well documented that the apoptotic processes promote neuroinflammation and neuronal degeneration in neurodegenerative pathologies such as Parkinson’s disease [23]. In the present study, we focused the search of proteins that are related to the establishment of inflammation response (Figure 5 and Table 2). Platelet-derived growth factor-D (PDGF-D) is a protein involved in leukocyte migration and, more interestingly, is involved in cell survival via ERK pathway and indirectly by the brain-derived neurotrophic factor (BDNF) pathway [24]. In addition to the overexpression of proteins involved in the response to oxidative stress, simvastatin administration induces at least a couple of other proteins, including the following: carcinoembryonic antigen-related cell adhesion molecule 1 (Ceacam1), a protein involved in the negative regulation of immune effector process [23], and sequestosome-1(Sqstm1), which is overexpressed in the simvastatin group. This protein is involved in the regulation of I-kappaB kinase/NF-kappaB signaling and involved in the regulation of apoptosis triggered by inflammatory cytokines [25].

Insulin-like growth factor I (Igf1) is a protein involved in the positive regulation of T cell proliferation. It positively regulates IL-4, Jun and Bcl-2 pathways, while negatively regulating TNF, IL-1β, Casp3, Bax, and NFκ-B1. This protein reduces its expression in rats that were exposed to MPP^+^ toxin [26]. 

TGF-β3 and TGF-β1 exhibited a similar profile expression. TGF-β1 is involved in the positive regulation of glial cell differentiation, and both are involved in the regulation of inflammatory response. Further, TGF-β1 has been reported to present a protective action against MPP+ injury in rat models [17,27]. MPP^+^ resulted in a reduction in TGF-β1 production in the substantia nigra and in primary VM neurons and microglia [25]. Allograft inflammatory factor 1 (Aif1) is involved in the positive regulation of leukocyte migration. S100A8 has a role in leukocyte aggregation. IL-1α is involved in the inflammatory response. Tumor necrosis factor ligand superfamily member 6 (Faslg) is involved in the death of T cells. IL-6r, a part of the receptor for interleukin 6, binds to IL6 with low affinity but does not transduce a signal. Signal activation depends on the association with IL6ST. Activation may lead to the regulation of the immune response and acute-phase reactions. IL-1β was detected in the Simvastatin and MPP^+^ groups; surprisingly, the higher expression was exhibited by the simvastatin groups. Caspase-6 is involved in the activation cascade of caspases responsible for apoptosis execution. This protein reduces its expression in the MPP^+^/Simvastatin group. Finally, IFN-α1 expression is reduced in the MPP^+^ group.

## 4. Materials and Methods

### 4.1. Ethics Statement

The protocol used in this study was approved by The Committee on Ethics and Use in Animal Experimentation of the Institute of Neurology and Neurosurgery and the standards of the National Institutes of Health of Mexico (Permit number INN-2017-2023). The study was performed following the Mexican regulations (NOM-062-ZOO-1999) and Guide for the Care and Use of Laboratory Animals of the National Institute of Health, 8th Edition to ensure compliance with the established international regulations and guidelines.

### 4.2. Animals and Treatments

Male Wistar rats (250–280 g) were housed in acrylic box cages and placed under constant conditions of temperature, humidity, and lighting (12 h light/dark cycles). Animals were provided with a standard commercial rat chow diet and water ad libitum. Experimental rats were administered on a subchronic schedule with a single dose—via the oral pathway (op)—of 40 mg/kg/day of simvastatin-tween 80 to 1% (Kendrick Pharmaceutical, Reg. No. 470M2002 SSA-IV), as vehicle or vehicle solution alone for seven days. Two hours after the last administration of simvastatin, the MPP^+^ was administered by means of an intracerebral surgery under ketamine/xylazine anesthesia (70/10 mg/kg). Animals were infused with MPP^+^ (10 μg/8 μL), or saline solution for the control group, into the right striatum with the following stereotaxic coordinates: 0.5 mm posterior, −3.0 mm lateral to the bregma, and −4.5 mm ventral [28].

### 4.3. Striatal Dopamine Levels Measurement

Twenty-four hours after the simvastatin administration schedule and MPP+ infusion, HPLC with electrochemical detection was used to measure striatal levels of dopamine (DA), as described previously [28]. Samples obtained 24 h after MPP^+^ injection were sonicated into 10 volumes of perchloric acid-sodium metabisulfite solution (1 M 0.1% *w/v*) and centrifuged at 10,000× *g* for 10 min before the supernatant was analyzed. Data were collected and processed via interpolation in a standard curve. Results are expressed as μmol of DA per gram of wet tissue.

### 4.4. Fluorescent Lipidic Products Assay

Fluorescent lipidic products are formed during the uncontrolled oxidation of polyunsaturated membrane lipids. They are considered to be composed of Schiff bases formed by the cross-linking of aldehydes from the oxidized lipids with surrounding amino acids [29]. Therefore, the increase in fluorescence signals in the extracted lipids are considered an estimation of oxidative damage.

The simvastatin effect was evaluated on the formation of striatal lipid fluorescent products (LPF) 24 h after administration of MPP^+^, as described previously [30]. The striatal tissue was homogenized in 2.2 mL of sterile saline, and 1 mL of the homogenate was then added to 4 mL of a chloroform-methanol mixture (2:1, *v/v*). The tubes were capped and vortexed for 10 s, and the mixture was then ice-cooled for 30 min to allow phase separation. The aqueous phase was discarded, 1 mL of chloroform layer was transferred into a quartz cuvette, and 150 μL of methanol was added. Fluorescence was measured in a PerkinElmer LS50B luminescence spectrometer at 370 nm of excitation and 430 nm of emission. The protein content was measured according to the method of Lowry et al. [31]. Results are expressed as arbitrary fluorescence units/μg protein.

### 4.5. Circling Behavior

Apomorphine-induced circling behavior was assessed in rats as previously described [29] and considered the end-point brain toxicity. Six days after MPP^+^ intrastriatal injection, animals were subcutaneously treated with apomorphine and ascorbic acid (1 mg/kg and 1 mg/kg, respectively), then placed into individual box cages. Five minutes later, the number of ipsilateral rotations to the lesioned striatum was recorded for 60 min. Rotations were considered as 360° turns, and the results were expressed as the total number of ipsilateral turns in a 1 h period (turns/h).

### 4.6. Immunofluorescence for Tyrosine Hydroxylase in the Striatum of the Rat

The effect of simvastatin on neurodegeneration induced-MPP^+^ in the rat striatum was measured by the presence of tyrosine hydroxylase (TH)—antibody dilution (1:100). Approximately 24 h after the behavioral evaluation, the animals (n = 3–4) were anesthetized with an overdose of sodium pentobarbital (80 mg/kg) and subjected to intracardiac perfusion with 200 mL of saline solution. The perfusion was followed by 200 mL of paraformaldehyde at 4%. The brains were removed, stored in 30% sucrose solution, and kept refrigerated until processing. Subsequently, 22 µm coronal cuts were made from the injured area (AP +1.6 mm to AP −0.48 mm related to Bregma) using a Leica brand cryostat, model CM 1520. The cuts were made in the injured area to carry out an optical percentage evaluation for the positive signal to TH.

Tissues were washed with PBS 3% (10 min), and antigen recovery was performed in citrate solution for 1 h. Three more washes were performed with PBS, and blocking was carried out with BSA (6 mg) and horse serum (4 µL) dissolved in 200 µL of PBS 3% for 1 h. Subsequently, the samples were incubated overnight with a mouse anti-TH antibody (1:100; sc-25269, Santa Cruz Biotechnology, Paso Robles, CA, USA.). Subsequently, three more washes were carried out with PBS-tween 3% and the samples were immediately incubated for 2 h with a secondary antibody (AffiniPure Goat Anti-Mouse IgG 1:100 Fluorescein (FITC)- AB_2338589 Jackson ImmunoResearch, West Baltimore Pike, PA, USA). Finally, the tissues were mounted on previously silanized slides with 3 µL of antioxidant solution. Nine slices of each brain were taken, in accordance with the coordinates already mentioned, from which photographs of six fields of the striatum of the injured hemisphere were taken for the determination of the TH signal. The photographs were processed as binary figures using Image J software to evaluate the percentage areas of both signals in each slide. Results were reported as the average area percent signal for TH.

### 4.7. Samples Protein Extraction 

Striatal tissue (St) from ST1 (Control), ST2 (simvastatin), ST3 (MPP+), and ST4 (simvastatin/MPP^+^) was washed three times with 1 mL of PBS, then homogenized for 1 min in the ice bath before the proteins were precipitated with acetone for 48 h; samples were centrifuged 6000 rpm/5 min/4° (3X); pellets were solubilized with 500 mL of 2D sample buffer (4% urea, 2% thiourea, 2% CHAPS, 160 mM DTT); then were precipitated with methanol-chloroform; pellets were solubilized with 200 mL of 2D sample buffer, centrifuged at maximum speed in microfuge 5 min at 4°, and supernatants were used for two-dimensional analysis.

### 4.8. Two-Dimensional-SDS-Page (2D-SDS-Page)

A total of 300 μg of Mc and St of all groups were individually loaded in IPG-strips of 7 cm with pH 3–10. After 16 h of passive rehydration, IEF was performed as follows: Step (1) 50 V/20 min/rapid; Step (2) 70 V/30 min/rapid; Step (3) 250 V/20 min/lineal; Step (4) 4000 V/2 h/lineal; Step (5) 4000 V/20000 Vh/rapid. Strips were treated with equilibration buffer with 2% (*w/v*) DTT (6 M urea, 2% SDS, 0.375 M TRIS-HCl pH 8.8, 20% glycerol and 1% bromophenol blue) for 15 min, and then with equilibration buffer with 2.5% (*w/v*) IAA for 20 min. Strips were loaded in precast 4–20% polyacrylamide gels and electrophoresis was run for approximately 35 min at 200 V. Gels were stained with Coomassie blue.

### 4.9. Bioinformatic Analysis

After Coomassie blue staining of the gels, they were scanned using ChemidocTM XRS Device (Bio-Rad Laboratories, Segrate, Milan, Italy) at 252 dpi resolution and analyzed using the PDQuest program. Spots that exhibited differential expression compared to the control were analyzed in the SWISS 2DPAGE public repository (https://world-2dpage.expasy.org/swiss-2dpage?pi_mw, accessed on 20 April 2023) to determine the identity of the proteins by comparison with MW and Ip using available 2D proteomic maps.

### 4.10. String Analysis

To determine the main signaling pathways involved in the simvastatin effect, we chose the proteins that exhibited differential expression between MPP^+^ and MPP^+^/Simvastatin groups for their analysis in the STRING database (https://version-11-5.string-db.org, accessed on 20 April 2023). The analysis showed that selected proteins (Figure 6) are biologically connected, exhibiting a PPI enrichment *p*-value of 7.64 × 10^−9^.

### 4.11. Statistical Analysis

Results from dopamine content and lipid fluorescent products were analyzed by one-way ANOVA followed by the Tukey test. Data obtained from evaluating the circling behavior were analyzed using the Kruskal–Wallis test followed by the Mann–Whitney test. Statistical significance was set at *p* < 0.05.

## 5. Conclusions

Simvastatin showed a significant antioxidant and, more importantly, neuroprotection effect against MPP^+^-induced neurotoxicity since it was able to counteract the oxidative damage exerted by MPP^+^, improving three damage markers: lipid peroxidation, dopamine content, and circling behavior. The first two were evaluated as short-term, and the last was considered a long-term endpoint damage marker. We also identified proteins responsible for inflammation response, which should be further studied.

## Figures and Tables

**Figure 1 ijms-24-10414-f001:**
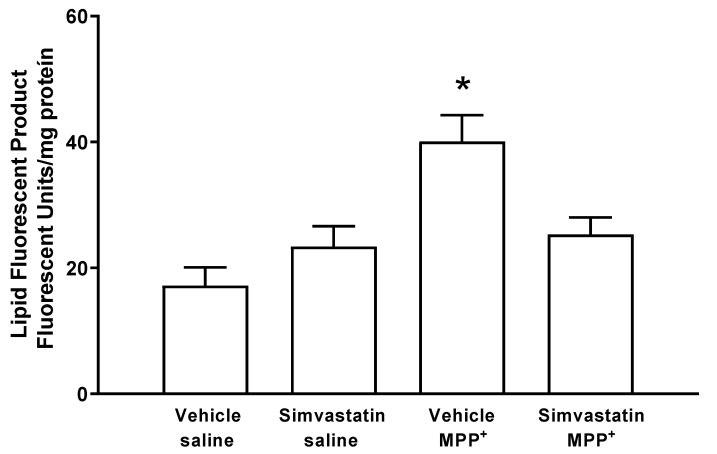
Simvastatin effect on the formation of lipidic fluorescent products in the model of neuronal damage by MPP^+^. Results are reported as mean ± S.E. There were 6–8 animals per group of fluorescent units per mg of protein. The differences were considered statistically significant from a * *p* < 0.05. All results were analyzed using a one-way ANOVA test followed by Tukey’s multiple comparison test.

**Figure 2 ijms-24-10414-f002:**
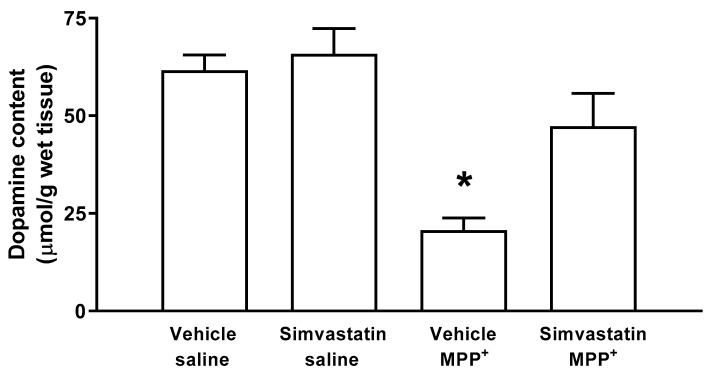
Effect of subchronic administration of simvastatin on the dopamine content detected by HPLC-DE. The results are expressed as mean ± S.E. There were 6–8 animals per group. Significant differences in DA concentrations between groups were considered statistically from a * *p* < 0.05. Data were analyzed with an ANOVA test followed by Tukey’s multiple comparison test.

**Figure 3 ijms-24-10414-f003:**
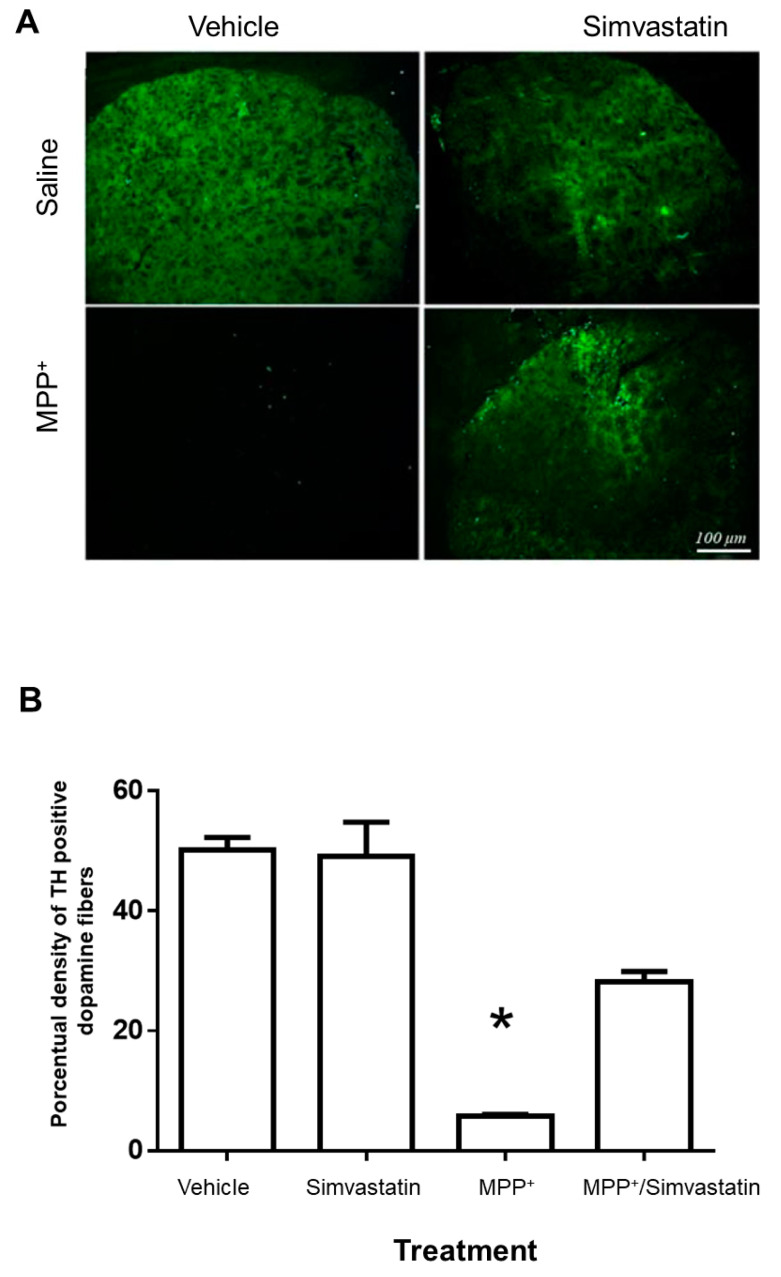
Effect of subchronic oral administration of 40 mg/kg/day of simvastatin on positive immunofluorescence to tyrosine hydroxylase (TH) in the rat striatum, enlarged images 20X). (**A**). The damage induced with MPP^+^ in the rat induced significant damage (* *p* = 0.001) relative to the control group, which was significantly decreased (*p* = 0.0053) in the group pretreated with simvastatin. Scale bar—100 μm. (**B**) Quantitative results of analyzed images are reported as the percentage of TH positive fibers ± S.E. of 5 animals per group.

**Figure 4 ijms-24-10414-f004:**
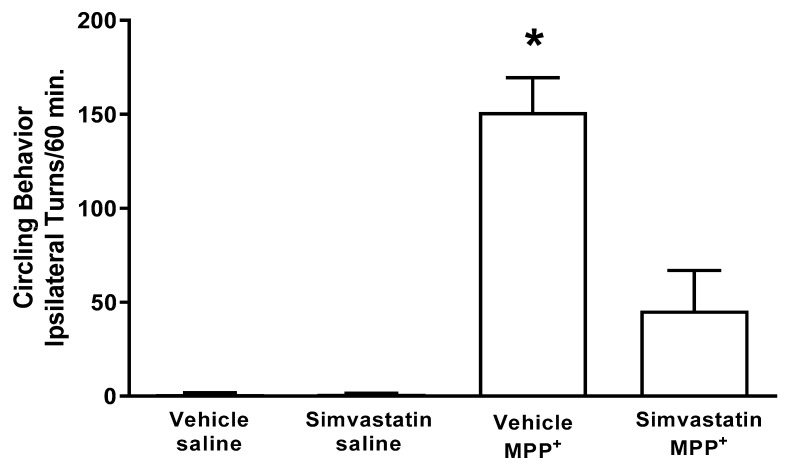
Effect of subchronic oral administration of 40 mg/kg/day of simvastatin on circling behavior induced by subcutaneous administration of apomorphine (1 mg/kg) 6 days after the intrastriatal administration of MPP^+^. The results are expressed as mean ± S.E. There were 6–8 animals per group. The differences were considered statistically significant from a * *p* < 0.05. Data were analyzed via a Kruskal–Wallis test followed by a Mann–Whitney U test.

**Figure 5 ijms-24-10414-f005:**
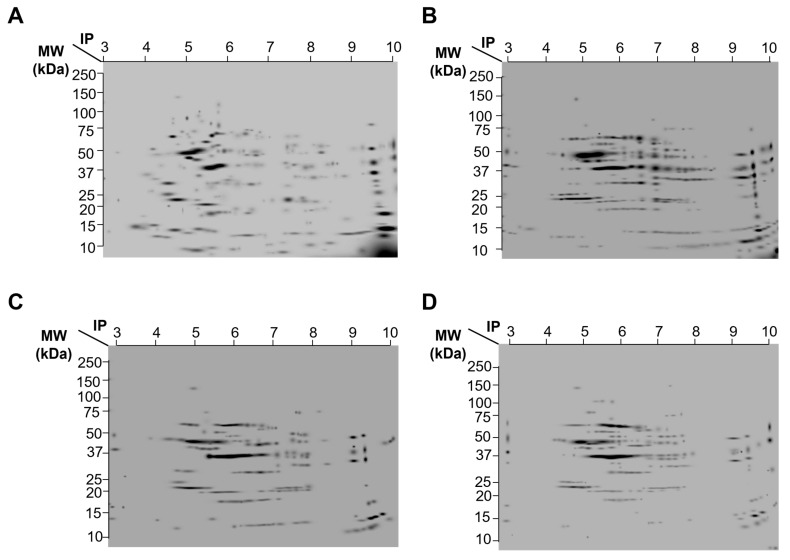
A composed image of the representative gels of 2D analysis of striated tissue of rat brain. (**A**) Control group without treatment. (**B**) Group treated with Simvastatin. (**C**) Group treated with MPP^+^. (**D**) Group treated with Simvastatin and MPP^+^. Whole extract of striated tissue of each group was separated by isoelectric point (pH 3–10 lineal, 7 cm strips) and molecular weight (4–20% polyacrylamide, precast gels).

**Figure 6 ijms-24-10414-f006:**
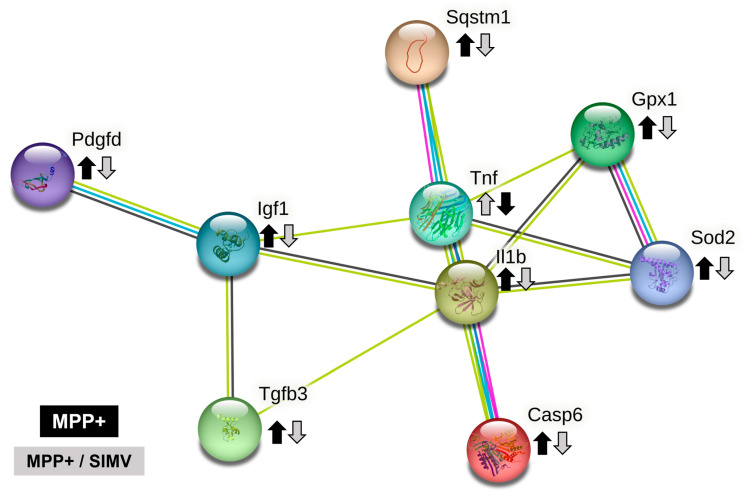
String network of proteins that are related to the inflammasome establishment. Comparison between MPP^+^ and MPP^+^/SIMV groups. We based the search on rat protein homologs as there is not a proteomic map of nigrostriatum of rats, and the impact of assuming a high homology among the identified proteins in other rat tissues and that the variations in molecular weight and isoelectric point would be minimal.

**Table 1 ijms-24-10414-t001:** Proteomic profile of striatum obtained from rats exposed to Simvastatin and MPP+ alone or in combination.

Group	Number of Spots	Number of Spots That Diminishes Their Expression	Number of Spots That Increases Their Expression
Control	224	-	-
Simvastatin	239	174	52
MPP^+^	177	194	32
MPP^+^/Simvastatin	130	211	13

**Table 2 ijms-24-10414-t002:** Proteins related to immune response that changed their expression with respect to the control. The use of ST1 (Control), ST2 (Simvastatin), ST3 (MPP^+^), and ST4 (MPP^+^- simvastatin).

SPOT	Protein Name	UniProtK B Accession Number	Theoretical Mr/pI	Experimental Mr/pI	Possible Function	Relative Expression			
						ST1	ST2	ST3	ST4
3501	Carcinoembryonic antigen-related cell adhesion molecule 1 (Ceacam1)	P16573	50.76/5.31	50.27/5.31	Cell adhesion, Positive regulation of activation-induced cell death of T cells.	+	++++	++	++++
5103	TGF-β2	Q07257	12.69/6.88	12.09/6.91	Cytokine	+	-	-	-
7403	TGF-β1 (Tgfb1)	P17246	44.33/8.59	40.94/8.52	Inflammatory response, Positive regulation of activation-induced cell death of T cells, Negative regulation of macrophage cytokine production.	++	+	-	-
4603	TGF-β3 (Tgfb3)	Q07258	47.12/6.10	48.11/6.06	Activation of MAPK activity.	++	++	+	-
6401	Proteinase-activated receptor 1	P26824	43.31/7.39	41.24/7.41	Inflammatory response.	+	-	-	-
4402	Heme oxygenase 1	P06762	33.00/6.08	33.57/6.09	Oxidoreductase, Apoptosis.	+	-	-	-
3302	IL-1α	P16598	30.86/5.59	30.32/5.53	Inflammatory response.	+	++	-	-
8301	IL-1β	Q63264	30.64/8.37	31.80/8.34	Inflammatory response.	-	++	+	-
3104	Interleukin-23 subunit alpha (IL-23α)	Q91Z84	19.60/5.63	17.92/5.62	Immunit, Inflammatory response.	+	-	-	-
3004	Protein S100-A8	P50115	10.10/5.69	9.46/5.64	Inflammatory response.	++	+	-	-
1302	Amyloid-beta A4 protein	P08592	30.03/4.15	26.92/4.13	Apoptosis, Cell adhesion, Endocytosis, Notch signaling pathway.	+	-	-	-
6206	Allograft inflammatory factor 1 (Aif1)	P55009	16.69/7.83	18.75/7.87	Cytoskeleton.	++	+	-	-
5302	B- and T-lymphocyte attenuator	Q6PNM1	31.11/6.77	30.35/6.77	Adaptive immunity.	+	-	-	-
3003	TYRO protein tyrosine kinase-binding protein	Q6 × 9T7	9.54/5.69	8.29/5.63	Immunity.	+	-	-	-
6203	IL-6r	P22273	24.36/7.77	22.37/7.66	Response to lipopolysaccharide.	+	++	-	-
5504	Corticotropin-releasing factor receptor 2	P47866	47.69/6.85	47.18/6.86	G-protein coupled receptor.	+	+	++++	++
2203	Leucine repeat adapter protein 25	Q566R4	18.71/4.95	18.97/4.95	Negative regulation of transforming growth factor beta receptor signaling pathway.	+	++	-	-
6503	Serine protease HTRA1	Q9QZK5	48.97/7.55	49.83/7.56	Negative regulation of transforming growth factor beta receptor signaling pathway.	+	-	-	-
9103	Insulin-like growth factor I (Igf1)	P08025	17.83/9.77	17.91/9.75	Growth factor.	++++	+++	++	-
2404	cAMP-dependent protein kinase type I-alpha regulatory subunit	P09456	43.09/5.28	43.18/5.27	cAMP, cAMP-binding, Nucleotide-binding.	+	++	++	-
8003	Caspase-8	Q9JHX4	10.83/9.13	11.21/9.13	Apoptosis.	+	-	-	-
3204	Myc box-dependent-interacting protein 1	O08839	25.23/5.72	24.85/5.73	Positive regulation of astrocyte.	+	-	-	-
2604	Myelin-associated glycoprotein	P07722	67.17/4.94	68.81/4.93	Axon regeneration.	+	-	-	-
4207	Matrilysin	P50280	18.93/6.22	19.35/6.21	Metalloprotease.	+	-	-	-
8102	WAP four-disulfide core domain protein 2	Q8CHN3	12.39/8.89	12.50/8.88	Serine protease inhibitor.	+	-	-	-
7002	Protein WFDC9	Q6IE41	6.39/8.19	6.78/8.20	Unknown.	+	-	-	-
6201	Vascular endothelial growth factor B	O35485	19.56/7.36	18.92/7.41	Angiogenesis.	+	-	-	-
2501	Calreticulin	P18418	46.34/4.34	46.96/4.34	Chaperone, Metal-binding, Zinc.	+	-	-	-
6101	Macrophage migration inhibitory factor	P30904	12.34/7.28	12.62/7.38	Inflammatory response.	+	-	-	-
2702	Cadherin-17	P55281	89.56/4.72	87.22/4.72	Cell adhesion.	+	-	-	-
3704	Protein artemis	Q5XIX3	78.18/5.63	79.45/5.62	DNA repair, Immunity.	+	-	-	-
6205	Interferon alpha-1 (Ifna1)	P05011	19.42/7.83	22.54/7.84	Cytokine pro inflamamtory.	+++	++	+	++
8104	Platelet-derived growth factor D (Pdgf-d)	Q9EQT1	13.96/9.39	13.22/9.35	Developmental protein, Growth factor, Mitogen.	++	-	+	-
8005	Tumor necrosis factor ligand superfamily member 6 (Faslg)	P36940	8.52/9.24	8.88/9.24	Proinflammatory.	++	+	-	-
2001	Protein S100-B	P04631	10.61/4.53	11.74/4.56	Positive regulation of myelination.	+	-	-	-
2705	Ubiquitin carboxyl-terminal hydrolase 10	Q3KR59	87.18/5.05	87.64/5.05	DNA repair.	+	-	-	-
2503	Sequestosome-1 (Sqstm1)	O08623	47.55/5.05	48.01/5.07	Apoptosis, Autophagy, Differentiation, Immunity.	+++	+++++	++	+
3201	Cdc42 effector protein 2	Q5PQP4	22.91/5.31	23.14/5.30	Rho protein signal transduction.	+	-	-	-
5204	Metalloproteinase inhibitor 4	P81556	22.55/6.88	22.76/6.87	Notch signaling pathway, central nervous system development.	+	-	-	-
5201	Caspase-6	O35397	18.06/6.48	18.10/6.49	Apoptosis.	++	+	+	-
5606	Synaptic functional regulator FMR1	Q80WE1	66.78/6.77	68.63/6.78	Translation regulation.	+	-	-	-
2607	Interleukin-2 receptor subunit beta	P26896	57.71/5.20	56.75/5.18	Cytokine pro inflamamtory.	+	-	-	-
6601	Ectonucleoside triphosphate diphosphohydrolase 1	P97687	57.40/7.49	63.42/7.45	Regulate purinergic neurotransmission.	++	+	-	-
2410	Tumor necrosis factor (TNF-α)	P16599	25.81/5.14	24.27/5.19	Cytokine pro inflammatory.	+	+++	++	++++

**Table 3 ijms-24-10414-t003:** Differential expression of antioxidant enzymes in striated tissue of rat brain. The use of ST1 (Control), ST2 (Simvastatin), ST3 (MPP^+^), and ST4 (MPP^+^- simvastatin).

SPOT	Protein Name	UniProtKB Accession Number	Theoretical Mr/pI	Experimental Mr/pI	Possible Function	Relative Expression			
						ST1	ST2	ST3	ST4
3306	Superoxide Dismutase 2 (SOD2)	P07895	15.91/5.89	19.55/5.88	Vasodilation.	-	++	+++	-
5705	Catalase (CAT)	P04762	59.63/7.15	58.07/7.20	Positive regulation of NF-kappaB transcription factor activity.	-	+	-	-
7203	Glutathione Peroxidase 1 (GPX1)	P04041	22.35/7.70	21.51/7.70	Angiogenesis.	++	+++	+	-

## Data Availability

The raw data supporting the conclusions of this manuscript will be made available by the authors, without undue reservation, to any qualified researcher.

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
