# Peer review of "Repurposing Simvastatin in Parkinson’s Disease Model: Protection Is throughout Modulation of the Neuro-Inflammatory Response in the Substantia nigra"

_ijms, 2023, doi:10.3390/ijms241310414_

Round 1

Reviewer 1 Report (New Reviewer)

The paper is all right, there are however, a few changes to be made:

page 2, line 61 - showed instead of induced

page 2, lines 72-74 - the phrase is incomplete, please rewrite

page 2,  line 75 - co-adjuvant

page 3, figure 1 - simvastatin + saline increased formation of L FP compared to control. Please comment…

page 6, line 178 - are you sure it's figure 1? 

page 16, line 313 - "there are no treatments"

page 16, line 314- "neuron half-life " ….

page 17, line 328- compared to lovastatin

pag 17, line 329  - Even while

pag 17, line 359 - that is because simvastatin has a clear effect . ..

page 21, line 541 - significant

Adequate. No extensive rewriting is necessary

Reviewer 2 Report (New Reviewer)

The entitled manuscript " Repurposing simvastatin in Parkinson's disease model: protection is throughout modulation of the neuro-inflammatory response in the substantia nigra" by Moisés Rubio-Osornio et al was aimed to the effect of simvastatin, which is a cholesterol-lowering drug, in other diseases, e.g. Parkinson's disease. The figures and results are convincing.

I was satisfied with the quality of the English language.

This manuscript is a resubmission of an earlier submission. The following is a list of the peer review reports and author responses from that submission.

Round 1

Reviewer 1 Report

Rubio-Osornio et al. presented the repurposing simvastatin in the rat model of Parkinson’s disease. This is an interesting study, suggesting that simvastatin could prevent striatal MMP+-induced damage and molecular mechanisms involved in this protective effect. There are, however, several issues to be addressed to further improve the manuscript.

1.     Figure 3 only shows the immunofluorescence images of TH in the rat striatum, while the results showing quantitative analysis of the percentage of TH-positive area is missing. Furthermore, a scale bar should be required in the image.

2.     Fluorescent lipidic products should be explained in more detail, specifically what are included.

3.     In tables 2 and 3, what do ST1,…and ST4 mean?

4.     In the discussion part, there are several sections that merely list the results.

Author Response

Rubio-Osornio et al. presented the repurposing simvastatin in the rat model of Parkinson’s disease. This is an interesting study, suggesting that simvastatin could prevent striatal MMP+-induced damage and molecular mechanisms involved in this protective effect. There are, however, several issues to be addressed to further improve the manuscript.

  1. Figure 3 only shows the immunofluorescence images of TH in the rat striatum, while the results showing quantitative analysis of the percentage of TH-positive area is missing. Furthermore, a scale bar should be required in the image.

The new Figure 3 has the scale bar now. Likewise, the figure caption was changed to be more accurate. Also, the quantitation was added as suggested.

  1. Fluorescent lipidic products should be explained in more detail, specifically what are included.

Answer. Fluorescent lipidic products are formed during uncontrolled oxidation of polyunsaturated membrane lipids, they are considered to be composed of Schiff bases formed by the cross-linking of aldehydes from the oxidized lipids with surrounding amino acids. Therefore, the increase of fluorescence signals in the extracted lipids are considered an estimation of oxidative damage. 

J. Triggs, L. J. Willmore (1984). In vivo lipid peroxidation in rat brain following intracortical Fe2+ injection. J Neurochem. 1984 Apr;42(4):976-80. doi: 10.1111/j.1471-4159.1984.tb12699.x.

In tables 2 and 3, what do ST1,…and ST4 mean?

We have included what the name of ST1, ST2, ST3 and ST4 means in the Table legend. (ST1 is Control), (ST2 is simvastatin), (ST3 is MPP+) and (ST4 stands for  simvastatin / MPP+). Also, in  4.7 section has been clarified (Samples protein extraction) section

  1. In the discussion part, there are several sections that merely list the results.

Indeed, however, they are being compared to other studies. We have modified accordingly. Thank you for your suggestion

We hope that our replies will satisfy reviewer concerns. We thank reviewer for the hard work put on reviewing our manuscript. With your help, a clearer and better draft of our manuscript was generated

Reviewer 2 Report

This manuscript presents data to support the repurposing simvastatin for the treatment of Parkinson's disease.   

It has several shortcomings:

1.     The dose of simvastatin used (40 mg/kg) in rats seems high. For a 70 kg human, this would scale to a dose of e 2.8 g. As well as being impractical, this very high dose is likely to cause debilitating side-effects.  

2.     The authors should refer to the ongoing clinical trial of simvastatin in people with secondary progressive multiple sclerosis. This study uses a dose of 80 mg/day.

3.     The authors may wish to measure the concentration of simvastatin in both blood and brain.

4.     Based on points 1-3, there seems little prospect of simvastatin being repurposed for the treatment of Parkinson's disease.   

5.     The data in the 3 tables have multiple comparisons, which increases the probability of producing false positive. Therefore, more stringent statistical analysis is required.

Author Response

This manuscript presents data to support the repurposing simvastatin for the treatment of Parkinson's disease.   

It has several shortcomings:

  1. The dose of simvastatin used (40 mg/kg) in rats seems high. For a 70 kg human, this would scale to a dose of e 2.8 g. As well as being impractical, this very high dose is likely to cause debilitating side-effects.  

Answer: The observation is pertinent if one thinks about of the possible administration of this statin to the human clinic. However, there are neuroprotection experimental designs in rats (pre-clinical models) that had administered simvastatin orally with a chronic frequency (3-4 weeks) at 20-50 mg/kg (Słupski et al., 2017; Vukšić et al., 2019). With based on these studies, we have established in this study the subchronic oral administration (1 week) of simvastatin of 40 mg/Kg. Undoubtedly, it will be of great interest in the near future to consider pharmacokinetic information for the dosing of this and other statins in preclinical and clinical studies. Furthermore, even when the dose of Simvastatin more frequently used to treat hyperlipidemia in rats range 10-20 mg/Kg, some other therapeutic applications require higher doses. For example, its use to promote osteogenesis requires doses ranging from 10-120 mg/Kg (Oryan A., Kamali A., Moshiri A. Potential Mechanisms and Applications of Statins on Osteogenesis: Current Modalities, Conflicts and Future Directions. J. Control. Release. 2015;215:12–24. doi: 10.1016/j.jconrel.2015.07.022).  Moreover, uses of Simvastatin as cardioprotective agent or to treat neuropathic pain needed a dose of 40 mg/Kg, (the same employed in the present study)

Simvastatin Protects Cardiomyocytes Against Endotoxin-induced Apoptosis and Up-regulates Survivin/NF-κB/p65 Expression.

Nežić L, Škrbić R, Amidžić L, Gajanin R, Kuča K, Jaćević V. Sci Rep. 2018 Oct 2;8(1):14652. doi: 10.1038/s41598-018-32376-4, Effect of simvastatin on sensorial, motor, and morphological parameters in sciatic nerve crush induced-neuropathic pain in rats.

Corso CR, Martins DF, Borges SC, Beltrame OC, Telles JEQ, Buttow NC, Werner MFP. Inflammopharmacology. 2018 Jun;26(3):793-804. doi: 10.1007/s10787-017-0425-1.

  1. The authors should refer to the ongoing clinical trial of simvastatin in people with secondary progressive multiple sclerosis. This study uses a dose of 80 mg/day.

Answer; The clinical trial referred is registered at the clinical trials.gov page: https://clinicaltrials.gov/ct2/show/NCT03896217 and cannot be used. The interest of this study was because they characterized the antioxidant and anti-inflammatory effects of subcronic administration of simvastatin in the Parkinson's disease. To support this, we used the neurotoxic model of MPP+. because, it is considered the model that reproduces the neurochemistry of human Parkinson's disease (oxidation effect). Reporting information on its adverse clinical effects in multiple sclerosis is not the objective. However, characterizing the adverse effects of simvastatin in other dopaminergic or non-dopaminergic pathways must undoubtedly relevant before applying any treatment to a controlled clinical trial of any neurological disease including Parkinson´s disease.

  1. The authors may wish to measure the concentration of simvastatin in both blood and brain.

Answer; The content of simvastatine and other statins measurement in different animal tissues and brain regions by analytical methods including HPLC is the objective of other preclinical studies. These measurements will be relevant in pharma and toxicokinetis of simvastatine administration in preclinical and clinical studies. Moreover, pharmacokinetics of Simvastatin, both in plasma and brain, have been already reported, after a dose of 40 mg/Kg in rats.

Al-Asmari AK, Ullah Z, Al Masoudi AS, Ahmad I. Simultaneous administration of fluoxetine and simvastatin ameliorates lipid profile, improves brain level of neurotransmitters, and increases bioavailability of simvastatin.J Exp Pharmacol. 2017 Apr 11;9:47-57. doi: 10.2147/JEP.S128696

  1. Based on points 1-3, there seems little prospect of simvastatin being repurposed for the treatment of Parkinson's disease. 

Answer; The antioxidant and anti-inflammatory results observed in this study allow the possibility of continuing the basic research of this and other statins (drug interactions, metabolic studies, and others) as well as similar molecules with the possibility of being considered in the future as an alternative in the clinic of neurological diseases that manifest oxidation and inflammation. as adverse mechanisms. The use of relatively high doses of Simvastatin for therapeutic uses is currently being studied to treat diseases other than hyperlipidemia. For example, in pneumonia (Simvastatin Improves Neutrophil Function and Clinical Outcomes in Pneumonia. A Pilot Randomized Controlled Clinical Trial.

Sapey E, Patel JM, Greenwood H, Walton GM, Grudzinska F, Parekh D, Mahida RY, Dancer RCA, Lugg ST, Howells PA, Hazeldine J, Newby P, Scott A, Nightingale P, Hill AT, Thickett DR. Am J Respir Crit Care Med. 2019 Nov 15;200(10):1282-1293. doi: 10.1164/rccm.201812-2328OC)

While some epidemiological studies show a significant effect of Simvastatin at high doses to prevent osteoporosis (Long-term effect of statins on the risk of new-onset osteoporosis: A nationwide population-based cohort study.

Lin TK, Chou P, Lin CH, Hung YJ, Jong GP. PLoS One. 2018 May 3;13(5):e0196713. doi: 10.1371/journal.pone.0196713).

Thus, it is possible that Simvastatin might be employed in future clinical trials to treat Parkinson´s disease. Alternatively, the use of doses of 10-30 mg/kg should be tested in the near future to find the dose-response of the anti-Parkinson effect of the statin, presented here.

  1. The data in the 3 tables have multiple comparisons, which increases the probability of producing false positive. Therefore, more stringent statistical analysis is required.

We compare Simvastatin, MPP+ and simvastatin / MPP+ groups against control by the PD Quest software in a qualitative analysis that resulted in the relative expression based in optical density based on the analysis of three independent 2D gels from each group.  So, there is no need of further statistical analysis, since results shows the level of each protein, and, their increase or decrease exprwssion level. There are not false positive, each spot it is what is shown, since the bioinformatic analyses matched the IP and MW on each protein found.

We hope that our replies will satisfy reviewer concerns. We thank reviewer for the hard work put on reviewing our manuscript. With your help, a clearer and better draft of our manuscript was generated

Round 2

Reviewer 1 Report

The authors put an effort in revising their manuscript and addressing issues raised previously. Given the nature of the work, several aspects still remain speculative. The authors’ future study should clarify these aspects.

Author Response

Thank you for your invaluable help

Reviewer 2 Report

II will restrict my response to the key issue with this manuscript: The dose of simvastatin used (40 mg/kg) is too high to be used safely in humans.

The author’s response is that similar doses have been used in other preclinical models of human disease. This is a very weak argument because it fails to relate preclinical data with clinical data.

In one study of simvastatin’s PK in humans (Br J Clin Pharmacol. 2009 May; 67: 520–526), administration of 40 mg of simvastatin produced a plasma Cmax of 1.9 ng/ml.

In rats, a dose of 40 mg/kg   produced a plasma concentration of 163 ng/ml  (J Exp Pharmacol 2017, 11;9:47-57), ie 86 times higher than that seen in humans following a 40 mg dose. In addition, the study of Al-Asmari indicates that simvastatin is metabolised by CYP3A4 so the potential for dangerous drug-drug interactions is very high. This risk, along with the known dose-related side effects of simvastatin, make the prospect o repurposing simvastatin to treat Parkinson’s disease untenable on the basis of the data presented.

Author Response

The reviewer is right, Cmax values for the dose employed here are higher than those attained in human beings. However, those concentrations are momentarily produced in plasma, as peak concentrations.

Perhaps the problem of the translational value of our results is better understood thinking that the MPP+ model of Parkinson´s disease is also produced with a single, high dose of the toxicant, while Parkinson´s disease is developed after years of neurodegeneration. Then, what is being modeled in the present study is the mechanism of neuronal death or neurotoxicity of MPP+ rather than Parkinson´s disease itself. This is now stated in the discussion section, as follows:

“Even when the results presented here are compelling, the translational value of the results are limited to explore the mechanisms of simvastatin neuroprotection against MPP+-induced neuronal dead. The dose employed here to prevent such damage produce higher plasma concentrations in rats (J Exp Pharmacol 2017, 11;9:47-57) than those attained in human studies at high doses (Br J Clin Pharmacol. 2009 May; 67: 520–526). Thus, one limitation of our work is the partial application to translational human studies.” 

Al-Asmari, Z. Ullah, A. S. Al Masoudi and I. Ahmad. Simultaneous administration of fluoxetine and simvastatin ameliorates lipid profile, improves brain level of neurotransmitters, and increases bioavailability of simvastatin’, J Exp Pharmacol, vol 11, no. 9, pp. 47-57, Apr. 2017, doi: 10.2147/JEP.S128696. eCollection 2017.

Krishna, A. Garg, B. Jin, S. S. Keshavarz, F. A. Bieberdorf, J. Chodakewitz and J. A. Wagner. Assessment of a pharmacokinetic and pharmacodynamic interaction between simvastatin and anacetrapib, a potent cholesteryl ester transfer protein (CETP) inhibitor, in healthy subjects’, Br J Clin Pharmacol, vol 67, no. 5, pp. 520-526, May. 2009, doi: 10.1111/j.1365-2009;2125. 2009. 03385.x. Epub 2009 Feb 4.